MPP-2025-37

# Asymptotic safety, quantum gravity, and the swampland: a conceptual assessment

Ivano Basile, [ORCID] $^{a,b}$ Benjamin Knorr, [ORCID] $^{c}$ Alessia Platania [ORCID] $^{d}$ and Marc Schiffer [ORCID] $^{e}$

$^a$ *Arnold-Sommerfeld Center for Theoretical Physics, Ludwig Maximilians Universität München, Theresienstraße 37, 80333 München, Germany*

$^b$ *Max-Planck-Institut für Physik (Werner-Heisenberg-Institut), Boltzmannstraße 8, 85748 Garching, Germany*

$^c$ *Institute for Theoretical Physics, Heidelberg University, Philosophenweg 12, 69120 Heidelberg, Germany*

$^d$ *Niels Bohr International Academy, The Niels Bohr Institute, Blegdamsvej 17, DK-2100 Copenhagen Ø, Denmark*

$^e$ *High Energy Physics Department, Institute for Mathematics, Astrophysics, and Particle Physics, Radboud University, Nijmegen, The Netherlands*

*E-mail:* `ibasile@mpp.mpg.de`, `knorr@thphys.uni-heidelberg.de`, `alessia.platania@nbi.ku.dk`, `marc.schiffer@ru.nl`

ABSTRACT: We provide a conceptual assessment of some aspects of fundamental quantum field theories of gravity in light of foundational aspects of the swampland program. On the one hand, asymptotically safe quantum gravity may provide a simple and predictive framework, thanks to a finite number of relevant parameters. On the other hand, a (sub-)set of intertwined swampland conjectures on the consistency of quantum gravity can be argued to be universal via effective field theory considerations. We answer whether some foundational features of these frameworks are compatible. This involves revisiting and refining several arguments (and loopholes) concerning the relation between field-theoretic descriptions of gravity and general swampland ideas. We identify the thermodynamics of black holes, spacetime topology change, and holography as the core aspects of this relation. We draw lessons on the features that a field theoretic description of gravity must (not) have to be consistent with fundamental principles underlying the swampland program, and on the universality of the latter.

# 1 Introduction

Formulating a theory of quantum gravity (QG) which consistently completes known low-energy physics is a formidable task, and remains one of the most daunting open problems in theoretical physics.[1] Gravity in our universe is very weak: even the heaviest of quarks has a mass $m_{\text{top}} \approx 10^{-17} M_{\text{Pl}}$. This means that, at the level of subatomic particles, low-energy gravitational physics is extremely weakly coupled. This state of affairs leaves one with little hope to directly probe QG effects with collider experiments in the foreseeable future. Without such guiding input on new physics, one is left with theoretical consistency. In this respect, at the level of non-gravitational particle physics, relativistic (Lagrangian) quantum field theory (QFT) has proven to be a spectacularly successful framework. The "rules of the game" involve very few building blocks, in the form of quantum fields and their interactions, constrained by consistency conditions such as unitarity, causality, symmetries, gauge redundancies and anomaly cancellation. As a result, at least at weak coupling, the low-energy description of physics is qualitatively universal, involving Yang-Mills gauge fields, fermions and scalar fields. The detailed structure (gauge group, representations, values of couplings, etc.) remains unfixed up to the above consistency conditions.

**Gravity as a QFT** — The successes of QFT outlined above naturally call for the question of whether gravity can be consistently described within this framework. Asymptotically safe quantum gravity (ASQG) [3–20] attempts to answer this question building on an interacting ultraviolet (UV) fixed point that makes the theory non-perturbatively renormalizable [21–62]. If the resulting UV critical surface has a finite dimension, physical quantities are predicted in terms of finitely many free parameters. Within some approximations schemes, this approach has been applied to the computation of couplings and masses in the standard model [59, 63–68], and in particular to Higgs physics [69–71]. Other proposed approaches for QG based on QFT include non-local quantum gravity [72] and quadratic gravity [73–75].

**Consistency as a driving principle** — Faced with the challenges of incorporating gravity into the QFT framework, it is important to build on solid grounds. At low energies, QG makes sense as an effective field theory (EFT), and from this starting point the problem of QG can be phrased as the search for a UV completion of such EFTs. This additional requirement further constrains the physics, and presents a path toward understanding QG on the grounds of its theoretical consistency. More optimistically, it may even eventually guide some experimental searches for new physics. These ideas led to the "swampland program".

**Swampland program as a bottom-up perspective on QG** — In the context of string theory, a number of patterns in the allowed range of EFTs were found and related to expected general features of gravity. These observations gave rise to proposals on the string landscape [76–78], which developed into a broader research avenue — namely the swampland program [79–81] — whose scope by now goes beyond string theory proper [82]. The essence of the swampland program is the search for physical principles which characterize QG from the "bottom up", namely without any explicit appeal to a specific microscopic theory or framework.

---

[1]See [1] for a recent overview and [2] for introductory lecture notes.

More precisely, swampland principles emphatically ought to be conditions which *cannot* be derived from the consistency of non-gravitational physics alone — a relativistic QFT may look perfectly consistent, but upon coupling it to gravity, it may develop subtle inconsistencies that relegate it to a mere EFT without gravitational UV completion. In this spirit, *within the scope of the present paper, we shall dub "quantum gravity" any (predictive) quantum theory which reduces to an EFT involving General Relativity (GR) at low energies and which is complete, namely it does not cease to be valid in any physical regime.*

**Towards extending the swampland question beyond string theory** — Within the swampland program, theories that become inconsistent when coupled to gravity are said to "belong to the swampland", as opposed to the "landscape" of gravitational EFTs which afford a consistent UV completion. Although the swampland program primarily originated and developed in the context of string theory, it sparked interest in the broader research community, and connections have been found in QG across approaches. Swampland criteria, such as the weak gravity conjecture [78, 83–87], the distance conjecture [77, 88, 89] and the no-global-symmetries conjecture [90–100] constrain gravitational EFTs with consistent UV completions, albeit unfortunately with a constraining power seemingly inversely proportional to the degree with which such criteria are established [79–81], see also section 2. As outlined above, whether such consistent theories are exclusively stringy is an open question, a positive answer to which is dubbed "string lamppost principle" [101–103]. Investigating these ideas beyond the context of string theory [82, 104–106] and comparing them to other top-down candidates would provide non-trivial tests of their validity, of string universality, or of the consistency of any given top-down candidate. In this paper, we make substantial steps in this research endeavor. We will restrict our discussion to QFT-based approaches, in particular ASQG, since they provide a direct connection to EFTs at low energies.[2] Our main focus will be *relating ASQG to a subset of swampland ideas, accounting for their logical (co)implications and the arguments substantiating them.* Nonetheless, the arguments presented in this paper, perhaps excluding those based on amplitudes as in section 4.1, apply to any QFT-based and Lorentz-invariant UV completion. We will mostly refer to string theory as a source of examples where such swampland ideas are realized, while motivating them *without referring to any specific UV completion.*

**Asymptotic safety and the swampland** — Some progress in defining and analyzing the asymptotic safety landscape and its interface with (some) swampland criteria has been made in recent years [82, 104–106, 121]. These investigations point to a non-trivial intersection between the asymptotic safety landscape and the space identified by some swampland bounds, namely the weak gravity conjecture [78, 83–87], the de Sitter conjecture [122, 123], and the trans-Planckian censorship conjecture [124, 125]. However, this approach has two limitations:

---

[2]We focus on spacetime dimensions $d > 3$. In $d \leq 3$ gravitons do not exist and gravity is topological, avoiding the arguments against a QFT description. In $d = 3$, gravity in anti-de Sitter (AdS) sectors can admit a Chern-Simons [107] or, more precisely, a Virasoro [108, 109] description as a field theory. In $d = 2$, Jackiw-Teitelboim gravity is UV complete but behaves as a statistical ensemble of theories [110–113], and accordingly lacks black-hole microstates. Variations of this theory afford a non-unitary worldsheet description [114–120].

first, merely testing certain bounds does not shed light on their bottom-up model-independent grounding, if any. Second, comparisons between bounds and top-down calculations in non-perturbative settings can be affected by uncertainties. In this paper, *we attempt to address these shortcomings with a qualitative investigation of the conceptual underpinnings of some swampland criteria, applying them to structural aspects of ASQG* which cannot be subject to systematic uncertainties. Assuming the consistency of ASQG, we will derive conclusions on its interplay with swampland ideas.

**The swampy starting point** — In order not to make implicit assumptions on the string lamppost principle, it is important to distinguish the swampland of inconsistent theories from the complement of the string landscape. Some work in this spirit appeared in [82, 105, 106]. As already remarked, none of our arguments will appeal to results from string theory except as examples of possibility. Rather, our reasoning will be based on foundational aspects of the swampland program, in particular spacetime topology change, black-hole thermodynamics, the nature of observables, and the interplay between information theory and the equivalence principle. These ideas, put together from literature spanning decades, provide a conceptual bottom-up grounding to the patterns observed in the string landscape, as well as reasons to apply certain swampland conditions to any given top-down candidate for a QG theory. We would like to stress at this point that the mole of swampland literature we will use for our arguments — which is probably opaque to most non-stringy readers — makes a pedagogical presentation of our ideas to the broad QG community very difficult. We will help the reader by explaining some of the concepts along the way, by repeating certain arguments multiple times, and by providing a dictionary at the end of the manuscript.

**The asymptotic safety starting point** — We shall focus on ASQG in the strict sense, i.e. the scenario in which QG is described by a *bona fide*[3] QFT, whose physical consistency for us includes principles such as (quantum) causality[4] and unitarity (of suitable observables) [129], as well as UV completeness. The latter necessitates the existence of a UV fixed point with a finite-dimensional critical surface in theory space. In addition, we shall restrict to Lorentz-invariant gravitational QFTs whose low-energy limit is dominated by GR (coupled to matter),[5] and we will work under the assumption that EFT remains valid in the presence of horizons [136–138]. The mere existence of a fundamental description in terms of quantum fields, together with the assumptions above, has non-trivial consequences for some of the arguments we shall present and examine. In particular, it entails a number of robust consequences which can be discussed beyond quantitative (but approximate) approaches involving func-

---

[3]We exclude theories with infinitely many species of quantum fields (except when they can be recast as higher-dimensional field theories) and string field theory. It would be interesting to investigate a limit of infinitely many (possibly higher-spin) species of fields in the context of ASQG, in order to capture topology fluctuations.

[4]In the asymptotic sense utilized e.g. in the context of the S-matrix bootstrap program [126–128].

[5]This excludes theories that yield strong infrared (IR) non-localities in the effective action [130–133]. Due to the Weinberg-Witten theorem [134], there are difficulties in realizing emergent gravitons as collective degrees of freedom. One option is a higher-dimensional field theory without gravitons, which produces them upon dimensional reduction [135].

tional renormalization [139] or dynamical triangulations [19]. Our discussion is also relevant in connection with the criticisms of [140].

**A note on "effective" ASQG** — When reflecting on the relationship between the string landscape and ASQG, it is noteworthy to highlight an alternative scenario to fundamental or "strict" ASQG, in which the fixed point marks the UV completion of the theory. Namely, one can conceive a scenario of "effective" asymptotic safety [104, 141–148], in which an intermediate regime between the IR and a non-field-theoretic UV is controlled by some (possibly non-unitary) fixed point, close to which renormalization group (RG) trajectories can be approximated by ASQG. Such a scenario would evade all of our ensuing considerations, which strictly rest on requiring field-theoretic physics at all scales. Outside the context of this paper, investigating the possible intersections (if any) between the string landscape, effective asymptotic safety and observational constraints may lead to fruitful links between these fields, as well as phenomenology. This is illustrated in figure 1. For our purposes, we would like to emphasize that the potential issues for a field-theoretic description of gravity raised by swampland arguments are wholly avoided in effective ASQG. If this scenario were realized, one would be able to approximate e.g. Wilson coefficients of consistent EFTs by those predicted by the RG flow stemming from a fixed point, any inconsistency of which would not impact the consistency of the full theory.[6] Such a setting would allow one to leverage the tools developed in the ASQG literature within an effective framework.

**Scope and findings** — Our aim is to draw lessons on the interplay between strict ASQG (or similar QFT-based approaches, as already stressed) and the most solid swampland ideas, that are as robust and rely on as few assumptions as possible. As a result of our analysis, we identify topology change and black-hole thermodynamics as the key features where strict ASQG in the above sense and certain aspects of the swampland program may clash. This entails that either the latter need not hold for consistency with physical principles, or that some of our assumptions on ASQG need to be relaxed. For instance, ASQG may need to display IR non-localities, or to be extended to include infinitely many fields, or to be realized at an effective level. Alternatively, black-hole thermodynamics would need a re-interpretation devoid of microstates.

**Paper content** — The contents of this paper are organized according to the overarching division between kinematic and dynamical considerations on QG. In section 2, we discuss some more detailed aspects of the swampland program and its consequences for EFTs, together with some examples. In section 3, we begin by discussing *kinematic aspects* of the swampland. By this, we mean any consideration that does not involve solving equations of motion, diagonalizing Hamiltonians, and any such dynamical ingredients. Rather, the kinematic structure of QG (at least in the EFT regime) consists of spacetime and gauge-bundle topology, (generalized) symmetries, and the types of fields and charges. In this context, we discuss how *completeness and the absence of global symmetries* arise from *spacetime topology*

---

[6]As an example, Planck-scale suppressed violations of positivity bounds in ASQG [105], if confirmed, could be interpreted as hints of this scenario being realized.

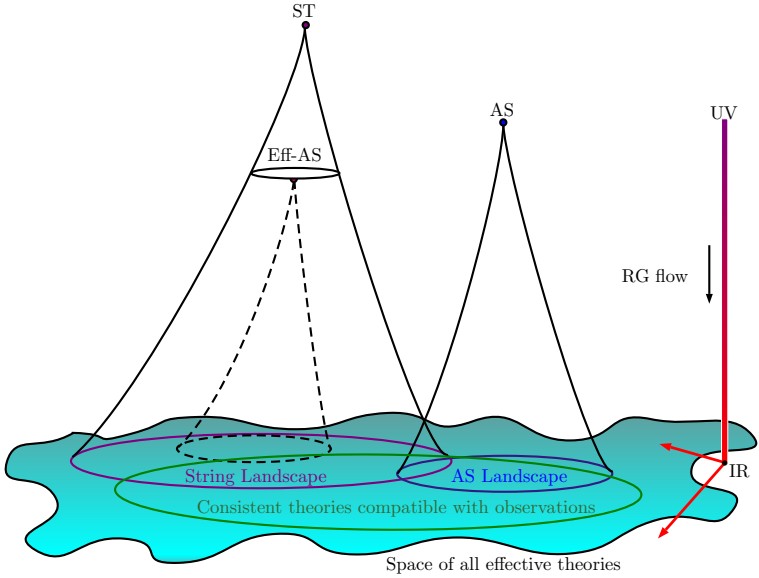

**Figure 1**. A schematic depiction of various logical possibilities regarding the string landscape and asymptotic safety landscapes. In its strict realization, ASQG is fundamental, and its landscape will presumably have a non-trivial intersection with the string landscape. If instead ASQG is a low-energy approximation of string theory, its landscape ought to fall within the string landscape. This paper discusses features of the asymptotic safety landscape from strict ASQG and their interface with a subset of swampland conjectures which are not necessarily tied to string theory, but can be motivated by arguments in EFT and black-hole physics.

*fluctuations, holography, and black-hole entropy*. We explain the logical connections between the various arguments and their consequences for ASQG. In section 4, we move on to *dynamical aspects*, which are less well-understood. We begin with the nature of observables in QG, their impact on weakly coupled UV completions and how ASQG can avoid it. Then we introduce the notions of *infinite-distance limits and towers of species*, their bottom-up motivations and the implications for field theories of gravity. As we shall see, such towers of species can only be consistent with ASQG in a particular case, and present some loopholes accordingly. Let us stress once more that our aim is to provide a general account of these ideas from a strictly bottom-up perspective, invoking string theory only to provide relevant examples. A middle-ground role in this sense is played by the anti-de Sitter/conformal field theory (AdS/CFT) correspondence, which can be viewed both in a top-down sense [149] and as a concrete general framework to study QG theories with stable, asymptotically AdS sectors, e.g. in the spirit of [92, 93, 150–156]. We shall conclude in section 5 with a summary of our arguments, conclusions, and loopholes. We explain some of the more technical aspects and terminology in appendix A. As already remarked, to make the life of the reader easier, throughout the manuscript the words and concepts explained in the dictionary are highlighted and linked to the corresponding explanation in the dictionary.

## 2 The swampland program and asymptotic safety

In the introduction we have outlined the general philosophy of the swampland program, focusing on its relation with QFT approaches to QG. Before delving into our main arguments in this regard, in this section we present some more technical aspects and results in this line of research. In particular, we highlight how swampland conditions can lead to severe constraints on classes of EFTs, which exclude almost all theories within (sometimes infinite) families.

**QG landscapes and string universality** — Whenever a relevant class of EFTs can be classified and its consistency assessed, the surviving subset of EFTs appears to be negligibly small with respect to the original set. It is conceivable that the swampland contains all but finitely many EFTs,[7] however many they may be. This line of research has been extensively pursued e.g. in [101, 102], exploiting the simplified structure of high-dimensional supergravities as testing grounds. Whenever the "swampland question" can be conclusively answered, it would appear that the surviving EFTs are in a one-to-one correspondence with string vacua. These results reinforce the idea of "string universality" [157], which has been complementarily analyzed via non-perturbative S-matrix bootstrap methods [158, 159]. However, at this level, the logical possibility that inequivalent QG theories give rise to (partially or fully) overlapping landscapes remains open [82, 106].

**Existence of landscapes and sparseness of vacua** — Generally speaking, it is not surprising that theories of gravity feature landscapes of solutions, and associated EFTs, in various (superselection[8]) sectors. Indeed, dynamical gravity allows for compactifications, which can produce multitudes of vacua. For example, the standard model EFT itself contains a landscape of lower-dimensional vacua [160, 161]. Of course, in this case we have the phenomenological advantage of being able to identify its unique four-dimensional sector as empirically relevant. As another example, string theory similarly and (in)famously features a landscape of different low-energy physical laws and constants, which can be considered vast for phenomenological purposes. However, in light of the preceding considerations, it is important to remark that it seems to be finite or, at worst, countable [101, 102, 162–171] — a far cry from the uncountable infinities of theory spaces[9] arising from quantum fields. An important lesson from the swampland program is that, if physically inequivalent theories of QG exist, the smallness of the string landscape is not a deficiency of string theory. Rather, it would be a shared trait of all potential landscapes, ultimately stemming from the unexpectedly restrictive consistency conditions due to gravity.

**Example applications of swampland constraints** — Let us provide a particularly sharp example of how swampland constraints can preclude QFTs from having a gravitational completion. The cancellation of gauge and gravitational anomalies of the traditional type is

---

[7]This is counting inequivalent EFTs at a fixed cutoff. For instance, if an EFT has a space of vacua parameterized by moduli (expectation values of gauge-invariant scalar operators), one would count such a moduli space as a single EFT insofar as the cutoff remains bounded.

[8]For example, at fixed number of infinitely extended dimensions.

[9]For example, varying the IR value of the fine-structure constant in quantum electrodynamics.

understood as a consistency requirement on (possibly gravitational) EFTs. Introducing QG, one is led to consider more general "Dai-Freed" anomalies [172],[10] which arise from spacetime topology change which cannot be ascribed to effective quantum fields (including graviton fluctuations). Including these, novel anomalies can appear. The standard model is remarkably devoid of them [172, 174], consistently with the existence of a gravitational completion.

Other constraints of this type stem from anomaly inflow. Namely, when the theory contains extended solitonic defects, the anomalous variation of the effective action under gauge transformation can acquire additional contributions localized on a defect. In order to cancel it, the defect itself must carry chiral degrees of freedom. The total anomaly then contains a second contribution from the degrees of freedom on the worldvolume of the defect. In the swampland context, the completeness of charge spectra [175] requires the presence of states with all possible gauge charges, which in some cases (such as high-dimensional supergravity) includes extended defects. The consistency of anomaly inflow then poses additional constraints which not all EFTs can satisfy. As an example, this requirement can rule out a gravitational UV completion for a number of supergravity theories in four [176], five [177, 178], six [175] and eight [179] dimensions.

The above results are based on anomaly cancellation. However, combining this condition with swampland conjectures arising from infinite-distance limits implies further bounds, e.g. that maximally supersymmetric $SU(N)$ gauge theory in four dimensions cannot be consistently coupled to (four-dimensional) gravity for $N > 23$ [180]. This bound is a consequence of the statistical properties of black objects and of the swampland distance conjecture, which we will review in section 4.2. The former is a more robust assumption but, as we shall discuss below, a consistent ASQG scenario would have to evade it. Thus, if such a loophole exists, the arguments of [180] would not exclude consistent theories with $N > 23$ in ASQG.

**Asymptotic safety and the swampland** — In the remainder of the paper, we will consider some basic ideas laying the foundations of the swampland program. While part of this line of research is inspired by string theory and patterns found therein, we stress once more that we focus on purely bottom-up arguments, and assume that these identify the universal landscape [106]. The spirit of this restriction is to reach conclusions that are as robust as possible when applying these ideas to ASQG and other QFT-based approaches. These ideas were pioneered in [82], where some swampland conditions were studied on the toy model landscape of one-loop quadratic gravity. The same strategy was subsequently followed in [105, 181], where positivity bounds were investigated in Einstein-Maxwell gravity to quartic order in derivatives. In this work, we are going to take a more conceptual approach, rather than focusing on computations in specific settings.

---

[10]The modern framework (see [173] for a recent review) describes anomalies of a $d$-dimensional theory $T$ in terms of a topological field theory $\mathcal{A}_T$ in $d + 1$ dimensions. Anomalies vanish if the partition function $\exp(2\pi i \mathcal{A}_T(Y)) = 1$ for all closed manifolds $Y$ with the appropriate structure for $T$ (spin, gauge bundle, etc.). Dai-Freed anomalies are those for which $Y$ is not a "mapping torus" $X \times [0, 1]/\sim$ over the spacetime $X$, where the two ends of the cylinder are identified with a gauge transformation [172].

# 3 The kinematic swampland

By and large, the best understood and developed aspects of the swampland program encompass the kinematics of QG [97]. The interplay between spacetime topology fluctuations [94, 96, 182–184], holography [185, 186], the absence of (generalized [187–192]) symmetries [90, 92, 93, 96, 193, 194] and completeness of gauge charge spectra [91, 195–197] form a tightly woven web of theoretical constraints, depicted in figure 2. How this web interfaces with approaches to QG other than string theory is an underexplored question, and in this section we are going to *analyze the compatibility between this web of kinematic swampland principles (cf. figure 2) and QG theories based on QFT in spacetime, such as ASQG. We will unveil a potential structural difference between the principles of the kinematic swampland conjectures and those of strictly field-theoretical descriptions QG; we will also clearly identify the root cause of such a difference.* As we shall see, the consistency of strict ASQG (cf. figure 1) or other QFT-based approaches seems to require a non-thermodynamic interpretation of semiclassical black-hole physics. This conclusion is supported by various lines of reasoning based on topology change and the thermodynamic behavior of black holes and related ideas, which we are going to motivate and discuss next. These considerations also interweave with the dynamical considerations that we shall present in section 4.1. We will also comment on loopholes and generalizations of ASQG that could avoid this conclusion.

## 3.1 Topology change in quantum gravity and field theory

The notion of summing over spacetime topologies, and whether it is required by consistency of QG, has been debated since its inception [182, 183]. Topology change lies at the heart of the kinematical swampland program (cf. figure 2), and hence its realizability in QG plays a key role in unraveling the interface between (kinematic) swampland conjectures and ASQG. For this reason, in the following we begin our discussion by reviewing some of the arguments that have been proposed which support topology change in QG, emphasizing their strong connections with black-hole thermodynamics, the ideas of no global symmetries [90], holography, and the completeness of charges. The key question is thus *whether (strict) ASQG and spacetime topology change are compatible with each other.* Here we provide arguments to the effect that, assuming that ASQG is consistent, its spacetime topology is fixed. This does not forbid spatial topology change over time. Rather, by spacetime topology change, we specifically refer to summing over spacetime topologies in the path integral. As emphasized in the introduction, our work brings together a substantial amount of technical literature, mostly coming from the string community, with the ideas at the core of ASQG. It is beyond our scope to review all the underlying results, but we will attempt to describe the intuitive reasoning and provide references for more details.

**Motivating topology change in QG** — We begin our discussion by examining the idea of spacetime topology change and its consequences for QG. Summing over spacetime topologies is an intriguing idea in the context of QG, dating back to the semiclassical considerations of [182, 183]. The notion of summing over spacetime manifolds in addition to

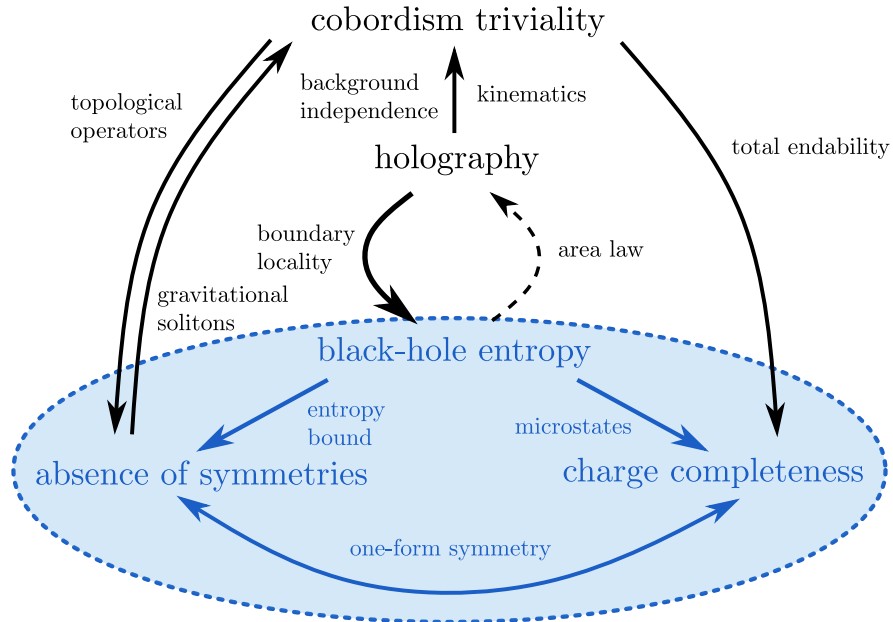

**Figure 2**. Various connections between the kinematic aspects of the swampland are shown. The dashed arrow denotes a weaker connection, indicating holography from the Bekenstein-Hawking area law. When specializing to AdS/CFT holography, one can prove [92, 93] completeness and the absence of global symmetries. The blue region in the lower part of the diagram contains connections which persist without summing over spacetime topologies. In appendix A we provide explanations for some of the technical jargon.

(geo)metric degrees of freedom defined on each given manifold is an analogous extension of how e.g. quantum Yang-Mills theory is constructed by summing over (isomorphism classes of) principal bundles in addition to connections. In other words, in Yang-Mills theory one must sum over the different topological sectors of the gauge field. In the latter case, including all bundles is crucial in order to preserve relativistic locality.[11] In gravity the traditional notion of locality is likely modified, but the approximate locality of long-distance physics is an indication that similar effects could play a role, although in the trivial spacetime topology the above considerations have to be restricted to exotic spheres [198]. More concretely, one could envision a violation of microcausal commutation relations for operators which are local relative to a gauge fixing. While such a procedure is impossible non-perturbatively even on the trivial topology [199], the perturbative BRST construction should be enough at low energies. It seems unlikely that such violations induced by excluding a suitable sum over gravitational instantons be consistent with S-matrix causality. Hence, *first motivations for topology change are the analogy with Yang-Mills theory and the compatibility with large-scale causality.*

---

[11]More precisely, one needs to preserve cluster decomposition of correlation functions.

More generally, the (semiclassical) path integral is usually considered to include families of metrics that degenerate to a different topology or become singular in some limit. Examples are given by the extremal limit of a family of charged black holes, or a family of increasingly pinched necks between two regions. Families of this type may arise by solving effective field equations, but generally appear off-shell in a path integral. Such boundaries in the space of metrics on a fixed manifold would likely be resolved to regular topological transitions by the UV completion. It would thus appear inconsistent not to sum over spacetime topologies, unless a peculiar suppression mechanism obstructs the transition. Hence, *a second motivation for topology change is the presence of singular boundaries between regions of the space of metrics in the path integral.*

The third argument is more technical. The sum over topologies can be argued to reflect a sort of redundancy of QG [97, 184, 200, 201], which can be understood in terms of the possible relation between gravity and entanglement in the sense of $ER = EPR$. In order for this to be possible, in the "gauge-fixed" description in which no explicit sum over topologies appears, the degrees of freedom of the theory must be able to replace its effects. A crucial ingredient in this respect is the triviality of cobordism classes [94], which also follows from the holographic principle [97] and subsumes the absence of (generalized) symmetries in gravity.[12] The absence of global symmetries can be independently motivated and related to completeness, as we shall explain in detail below. In summary, *the third motivation for topology change is the $ER = EPR$ correspondence and the holographic principle.*

**Topology change in a QFT of gravity** — Quantum fields are defined on the spacetime manifold.[13] Thus, it seems structurally impossible to reproduce the effects of topology change, like gravitational solitons [96, 97], within a strictly field-theoretic framework. Additional support for such obstructions comes from the difficulties, both practical and conceptual, in implementing a sum over topologies within a putative complete functional integral: even if the unwieldy and unenumerable sum over all diffeomorphism classes [205, 206] of (Riemannian) manifolds were restricted,[14] in the absence of a complete set of topological invariants one would need to arbitrarily pick out representative manifolds within the restricted class. Moreover, the global obstructions of gauge fixing [199] in terms of the Gribov problem, and to write down flow equations for generic topology, render a continuous path integral approach in terms

---

[12]To see this, on the one hand one can observe that non-trivial cobordism classes $[M] \in \Omega_k$ would provide conserved charges under a $(d-k-1)$-form symmetry [187] group $\Omega_k^\vee$, which are physically carried by gravitational solitons [96] kinematically described as connected sums $\mathbb{R}^{d-k-1,1} \times (\mathbb{R}^k \# M)$. On the other hand, such objects break (and abelianize) non-invertible symmetries, and provide the necessary representations to achieve gauge charge completeness [96].

[13]More precisely, they are operator-valued distributional sections of associated bundles to the frame (or an equivariant lift thereof, such as spin) bundle over spacetime. Functorial formulations [202–204] also show the dependence on a spacetime manifold.

[14]For instance, in [184] it is argued that the allowed topological transitions are described by surgeries, which leave the cobordism class invariant. In general, within ASQG there cannot exist microscopic defects killing nontrivial classes, because they would constitute non-field theoretical degrees of freedom [94, 97]. Thus, generically the sum over surgeries from a given manifold does not cover all topologies.

of quantum fields problematic. For instance, while local curvature terms in an early-time expansion of the heat kernel do not depend on subtleties of this type, the global topology affects other contributions, such as Kaluza-Klein modes on tori relative to Euclidean space.[15]

As a final comment, let us emphasize that among the various topological invariants characterizing the spacetime manifold within a functional integral or flow equation, in four dimensions the only ones that can be written as the integral of a local quantity are the Gauss-Bonnet (or Euler) term $\int R \wedge \star R$ and the Pontryagin term $\int R \wedge R$. In current implementations [52, 216], their couplings cannot attain non-trivial fixed points. Hence, if topology change were somehow allowed, this would lead to infinite weights for different topologies, in particular those provided by gravitational instantons.

To summarize, *topology change is motivated by an extensive network of arguments, but they seem very hard to implement in a QFT of gravity on spacetime*, since quantum fields live, by definition, on spacetimes of fixed topology. In the absence of topology change, the network of swampland ideas in figure 2 is left with the blue-shaded region: black-hole thermodynamics, completeness of charges, and the absence of global symmetries — which we are going to discuss next — are still required by consistency based on semiclassical arguments.

## 3.2   Global symmetries and semiclassical black holes

Although the arguments based on holography [92, 93, 97, 99, 100, 217] are perhaps the most solid, the original and most intuitive understanding behind the notion that global symmetries be absent in QG is grounded in semiclassical black-hole physics [90]. Various versions of these arguments have been discussed and refined informally, but they all start from the observation that in the semiclassical limit, Hawking radiation is blind to global charges. This is in stark contrast to gauge charges, which can be measured or defined by fluxes at infinity, and indeed do characterize the spectrum of black holes. Schematically, considering a large black hole, one can throw global charge inside and wait for the evaporation to bring it back to its original mass. On average, the emitted radiation contains no global charge, and thus the EFT would describe the resulting black hole as degenerate with the original one. For continuous groups with infinitely many possible charges, this results in a blatant violation of the (markedly finite [218, 219]) Bekenstein-Hawking entropy scaling.

Other variants of this argument involve allowing the black hole to evaporate up to the point at which it would kinematically not be able to keep radiating without shedding global charge (due to e.g. a mass gap in the charged spectrum), thus leaving a remnant which can lie

---

[15]There are examples where the effect of summing over topologies can be reliably computed, either within EFT or a UV completion [207–214]. In particular, [211, 212] reproduces the Bekenstein-Hawking entropy by effectively counting an overcomplete set of states available in the low-energy description, similarly to how the appropriate non-trivial thermal saddle does in the semiclassical path integral. In some cases, these microstates can be counted in string theory (starting from the seminal work of [215]), and under the conventional interpretation of black-hole entropy it is compulsory that the results match. The latter examples arise in string theory [208–210, 213], where the sum over topologies is encoded in the fact that the relevant partition function only depends on the asymptotic boundary of the noncompact spacetime. The redundancy described in [97, 200, 201] in this context emerges from the stringy degrees of freedom present in the theory.

well within the regime of validity of the effective low-energy description, if the starting mass and/or charge is large enough. The resulting large remnants would be in conflict with the behavior of Hawking radiation in the same regime. Moreover, even Planckian remnants have been argued to lead to thermodynamic pathologies due to their overproduction [220].[16]

Yet another *a priori* independent way to reach the same conclusion is to use charged black holes in gauge theory to show that the spectrum of gauge charges must be complete, and thus (non-invertible) symmetries must be broken [96, 193, 194]. In order to arrive at this conclusion, one can observe that the spectrum of black holes accessible to the EFT involves any parametrically large charge $Q \gg 1$ and thus also $Q \pm 1$ for any such $Q$. The statistical interpretation of black-hole entropy then implies that a consistent UV completion must contain states of any charge. We consider discrete charges since the gauge group should be compact, otherwise one runs into violations of entropy bounds [91].

## 3.3 Consequences of no topology change

In section 3.1 we argued that the strictly field-theoretic formulation of ASQG and spacetime topology change are hardly compatible. We shall now examine some consequences of this possibility, which more generally ought to apply to any QG theory without topology change. In the absence of topology change, the network of intertwined principles depicted in figure 2 reduces to the blue-shaded region. The absence of topology change thereby has consequences for the arguments leading to the absence of global symmetries, charge completeness, and black-hole entropy that we discuss subsequently.

**Symmetries and completeness** — In the above discussion on topology change, we have emphasized the role of global symmetries (or lack thereof), as well as their relation to the completeness of gauge charge spectra and cobordism triviality. As we have remarked, the latter connection can be understood as a consequence of the holographic principle, and in particular the concrete framework of AdS/CFT duality allows proving that no global symmetries exist [92, 93][17] and charge spectra are complete [196, 197]. Our preceding arguments show that in ASQG there should be no sum over topologies, and thus no holography.

In the absence of spacetime topology change, the most robust arguments to deduce the absence of any and all global symmetries,[18] completeness of gauge charge spectra [91, 195] and their equivalence [96, 193, 194] due to gravitational solitons [96], and cobordism triviality [94, 222] are lost. Dynamically, contributions to processes violating global charge conserva-

---

[16]A similar reasoning for weakly broken symmetries leads to the so-called "bad breaking" condition, stipulating that Ward identities ought to be violated by an $\mathcal{O}(1)$ amount when probing the theory at or above the cutoff [87].

[17]Similarly, in string perturbation theory no continuous global symmetries exist [91], since any symmetry on the worldsheet is gauged in spacetime. This result is believed to extend to discrete settings, but it is conceivable that it may not be visible perturbatively. On the other hand, the spacetime fate of non-invertible worldsheet symmetries is a peculiar type of Higgsing [221] which reflects the effects of gravitational solitons described kinematically in [96].

[18]Be it group-like, higher-form, categorical, non-invertible — perhaps the most general such notion is embodied by a (higher) fusion category of topological operators, up to the finiteness axiom for simple objects.

tion [99] and black-hole entropy [211, 212, 223] due to non-trivial topologies are absent. Does this mean that global symmetries are allowed and gauge charge spectra may not be complete if ASQG is consistent? As we discussed in section 3.2, *one can argue for either constraint purely on grounds of semiclassical black-hole thermodynamics* [90], without invoking holography, topology change, or UV physics (except for unitarity). However, without the solid interconnected network of logical implications provided by topology fluctuations, see the blue-shaded region in figure 2, *the resulting case is less compelling than in the full picture*. However, as we shall see in the next point, black-hole thermodynamics needs a different interpretation, and thus the close relationship with global symmetries is lost.

The potential presence of global symmetries in ASQG is supported by how RG flows interact with Ward identities when the regulator preserves a non-anomalous symmetry. More precisely, Ward identities show that no symmetry-breaking terms are generated by the flow if the regulator does not break the symmetry [224]. This has been confirmed in various explicit computations for global symmetries of kinetic terms. For example, shift symmetry [225–228], chiral symmetry [142, 229–232], or $O(N)$-symmetry [233–236] are not broken by gravitational fluctuations. This means that, first, all explored symmetry-breaking terms can be consistently set to zero at all scales, and second, that the symmetric subspace has an interacting UV fixed point, see also [8, 236] for a discussion.[19] There is also a first piece of evidence from lattice computations that a global shift symmetry of quenched scalars is unbroken [237]. Another observation related to (the lack of) symmetries is that the pattern of generalized $p$-form symmetries extends down to $p = -1$, where a $(-1)$-form "symmetry" encodes a free parameter/coupling in the theory [95]. Since UV-critical surfaces typically have non-zero dimensions, it would be interesting to assess the role of these "symmetries" in ASQG.

**Holography and black-hole entropy** — Relinquishing topology change requires abandoning holography as well, since the latter implies the former [97]. Intuitively, the reason is that localized excitations in distant regions of the boundary can be in an entangled state or not. Such entangled states are holographically dual to a wormhole connecting the localized regions on the spacetime boundary. Conversely, non-entangled states correspond to spacetimes without wormholes connecting those boundary regions. Since all quantum superpositions are allowed, different topologies are related to holography via wormholes. Also, summing over wormholes in the EFT recovers the Bekenstein-Hawking area law [211, 212], indicating boundary locality from the bottom up. Stipulating that these wormhole configurations be excluded in the functional integral also eliminates the most direct derivations of black-hole thermodynamics (namely, the ones in [211, 212, 223], since they rely on summing over topologies), whereas the one based on quantum fields on black-hole backgrounds [238] still applies. The latter derivation implies that there is a thermal behavior of the quantum fields themselves — presumably in (quasi-)thermal equilibrium with the black hole. This raises a natural question: if quantum fields (including gravitons) in the background of a (non-extremal) black

---

[19]This does not mean that, in general, all fixed points admit global symmetries, or that one could not break it dynamically even if one starts at a fixed point with a global symmetry.

hole feature a thermal behavior of a type that cannot be globally ascribed to an accelerated reference frame (such as in the case of the Unruh effect), what are these degrees of freedom in (quasi-)thermal equilibrium *with*? Moreover, what mechanism satisfies the second law of thermodynamics when throwing information into a black hole, if the corresponding area increase cannot be interpreted in this fashion? These apparent contradictions — implied by the absence of topology change in gravitational QFTs — are the first indication that the standard interpretation of black-hole thermodynamics may be modified in a consistent ASQG scenario. As a second complementary indication, in the absence of topology change, spacetime fields cannot reproduce the non-localizable super-exponential density of black-hole microstates [239]. Finally, a third indication comes from holography. Since the latter is also hinted at by the Bekenstein-Hawking scaling of the (von Neumann) entropy

$$S_{\mathrm{BH}} \overset{A \gg \ell_{\mathrm{Pl}}^2}{\sim} \frac{A}{4\ell_{\mathrm{Pl}}^2} + \mathrm{const.} \times \log \frac{A}{\ell_{\mathrm{Pl}}^2} \tag{3.1}$$

by virtue of the difficulty in reproducing an area law for a mixed state which is available in the EFT without local boundary dynamics (see also [240]),[20] the epistemically most economical route for ASQG seems to involve a significant alteration in how one should interpret black-hole thermodynamics. A popular but somewhat superficial argument [140] states that the presence of a UV fixed point implies that the density of high-energy states behaves like that of a conformal field theory in the bulk (thus scaling as a volume, rather than as an area). Potential loopholes of this argument [6, 242] include large anomalous scalings, as well as the option of realizing scale-invariant QFTs that are not conformally invariant [243], although, in the relativistic setting, no interacting example has been found so far in four dimensions and above. Alternatively, one could try giving up IR locality (which might also be relevant in relation to avoiding global symmetries [133]), although it may bring along other issues, related to additional degrees of freedom, Wick rotation, and observations. The conclusion that *black-hole thermodynamics needs a reinterpretation in UV-complete gravitational QFTs* is one of the main upshots of our analysis, and we will further corroborate it in section 4.

This is perhaps not surprising — after all, the general picture motivating ideas beyond field theory is that gravity is a coarse-grained, thermodynamic, emergent description. In a purely field-theoretic approach for the gravitational field,[21] this cannot be the case.

It is therefore apparent that, while these arguments do not appear to rely on holography, at least directly, they do at least rely on the standard statistical/thermodynamic understanding of semiclassical black-hole physics. Once more, we are led to conclude that, in order to avoid inconsistencies between black-hole thermodynamics and ASQG, one must stipulate

---

[20]In consistent QG theories that include an asymptotically AdS sector, the connection between the area law and holographic dynamics is starkly visible via the AdS/CFT correspondence [149, 241], where (non-extremal) black holes are encoded in *bona fide* (thermal) states of a local conformal field theory. In this sense, the (grand-)canonical partition function computed within the EFT is, in fact, an actual thermodynamic quantity.

[21]A field-theoretic approach in which the gravitational field is composite, rather than fundamental, would seem to be in conflict with the Weinberg-Witten theorem, since any source of such compositeness would presumably carry stress-energy.

that the latter cannot be interpreted in the standard thermodynamic and statistical fashion. In section 4, we shall discuss another independent motivation, grounded in the *asymptotic darkness* of observables, pointing to the same conclusion.

All in all, excluding topology change leaves ASQG to address arguments based on black-hole physics, in particular entropy considerations, which in turn are strictly related to completeness and the lack of global symmetries. Presumably, one way out that acknowledges at least the latter two conditions is simply to state that ASQG is to be supplemented by additional conditions, much like e.g. anomaly cancellation in gauge theories.

## 3.4 Generalizing ASQG beyond QFT

Is there a generalization of the standard framework(s) of QFT which can get around these issues while still remaining within the framework of ASQG? It seems difficult to answer this question without an extended definition of these terms. Perhaps a seed of this idea is contained in some of the technical approaches used to perform computations in this context, namely those based on the functional RG. In this context, flow equations can define a theory without direct reference to a functional integral over quantum fields. This in principle avoids issues in directly defining functional integrals for gravity, such as the absence of a non-dynamical physical flowing scale. In other words, one may not need coordinates on the space of metrics if one could define observables purely via an auxiliary flow equation. However, this approach still cannot incorporate a sum over topologies directly,[22] nor indirectly, since there cannot be any non-field-theoretic degrees of freedom that could replace its effects. In addition, it seems unclear how to encode the physics of extended operators (such as spin holonomies [244]) and other global effects (like Dai-Freed anomalies [172]) required by topology fluctuations (see e.g. [94, 96]) without formulating a more general functional formalism. In a scenario in which such a structure were used to describe something qualitatively different from a functional integral with quantum fields over spacetime, it seems difficult that it would reproduce the semiclassical functional integral which describes the low-energy EFT. However, matrix and tensor models could provide such a framework in which spacetime and topology change emerge along with quantum fields. Specifically, tensor models are also interesting because some may be asymptotically safe [236, 245–249], and some are related to string theory [250–255].

## 3.5 Summary and loopholes

The arguments for kinematic swampland constraints that we discussed hinge on spacetime topology change or the thermodynamics of semiclassical black holes, in particular the former implying the latter. Either of these ingredients is crucial to support the absence of global symmetries and the completeness of charge spectra (see figure 2). In the context of ASQG, we argue that the sum over spacetime topologies is very difficult to implement, unless one stretches the definition of QFT, e.g., by including infinitely many fields. Even without such

---

[22]If it could, it would still hardly be related to QFT as we know it, even in its most abstract formulation, because it would lose the functorial assignment of Hilbert spaces, unitary operators and partition functions to manifolds due to the sum over topologies.

a sum, the considerations based on black-hole entropy, i.e. the asymptotic area law and the existence of microstates (of any possible gauge charge), still apply. In order to circumvent the issues in explaining them with a fundamental QFT of the gravitational field, we can consider a few possible loopholes:

- the thermodynamic interpretation of black holes is somehow modified;

- the theory features strong IR non-localities overpowering the dominance of GR at large distances (see also [132, 133]). This may also lead to causality violation or tensions with observations, which we do not discuss in detail;

- QG violates our working definition: that of a Lorentz-invariant, unitary (and causal) quantum theory which reduces to EFTs dominated by GR at low energies;

- ASQG is not fundamental, e.g., it is realized in the form of effective asymptotic safety [104, 141–148], or it must be modified to feature infinitely many fields.

With this summary on the kinematic aspects of the swampland, we now move on to the dynamical aspects in the following section.

## 4   The dynamical swampland

We now turn to dynamical aspects of the swampland and their relation to ASQG. Perhaps as expected, the dynamical aspects of QG are comparatively less clear than the kinematic ones discussed in the preceding section. Important exceptions comprise settings in which gravity is weakly coupled,[23] where a number of available results from S-matrix bootstrap methods show an incompatibility between unitary and causal graviton scattering on the one hand, and a strict field theory formulation due to infinite towers of higher-spin species on the other. These ideas are further reinforced by complementary arguments based on black-hole physics [256–260]. Namely, in weakly coupled QG, the smallest black holes in the EFT have size $\Lambda_{\rm EFT}^{-1} \gg \ell_{\rm Pl}$, and they transition to (an ensemble of) species below this threshold [260]. Matching thermodynamic quantities [257, 260] and scattering amplitudes [259] at the transition constrains the density of states of these infinite towers of species. These considerations do not lead to an intrinsic issue for the consistency of ASQG, since any such scenario must be non-perturbative anyway. However, they will become important in section 4.2, where weakly coupled limits can appear at infinite distance in theory space. For the time being, we first focus on the role of observables in QG in order to further support the claims made in the preceding section. Then, in section 4.2 we discuss the consequences of information-theoretic factorization in gravity along infinite-distance limits.

---

[23]Weakly coupled in this context means specifically that the UV cutoff $\Lambda_{\rm EFT} \ll M_{\rm Pl}$ be parametrically sub-Planckian. This is because, within EFT, graviton couplings are bounded by $\Lambda_{\rm EFT}^2/M_{\rm Pl}^2 \ll 1$. Conversely, a theory with $\Lambda_{\rm EFT} = M_{\rm Pl}$ is called strongly coupled. In this precise sense, ASQG is likely strongly coupled.

## 4.1 Observables and UV/IR mixing

What are the observables in QG? No local quantities are gauge invariant, and there are no probes because of the universal coupling to gravity which makes everything interact with everything.[24] Global observables may be (possibly relational) spacetime integrals of local quantities, or they may be holographic,[25] i.e. defined purely by data at the asymptotic boundary. In asymptotically AdS superselection sectors, the latter are boundary correlators, as in the AdS/CFT correspondence [241] where they arise from a dual conformal field theory; in the asymptotically flat case, they are given by S-matrix elements. It is not clear how this story fits with cosmological de Sitter(-like) settings [125, 261–266], although perturbatively observable algebras may be defined on observer's worldlines [267, 268].

**Scattering amplitudes as sharp observables** — Since asymptotic boundary data is fixed, if the sum over topologies is included in the theory, observables can only depend on boundary data (and free parameters). Thus, when topology fluctuates, relational spacetime integrals cease to be well-defined. Spacetime integrals of local quantities must be defined for each topology, and summed over, resulting in a boundary observable. More optimistically, if the path integral could be somehow adapted to include topology fluctuations, it would cease to depend on the bulk spacetime manifold and become holographic due to the sum over topologies, retaining only its dependence on boundary data. Hence, in such a theory, one expects any observable to be holographic at least in the above sense. Conversely, the holographic principle implies a sum over topologies as discussed in [97]. All in all, independently of the sum over topologies and focusing on asymptotically flat sectors for concreteness, scattering amplitudes are always sharp observables, independently of whether or not global observables are sharp, and they can be investigated in a bottom-up fashion via bootstrap methods [269]. A recent discussion on how this framework can connect to approximately local physics can be found e.g. in [270]. An alternative operational understanding of why boundary observables are a deep feature of QG is tied to the properties of black holes. Namely, as discussed e.g. in [271] (see the interview with Arkani-Hamed), any attempt to define a quantum observable in a local fashion using a measurement apparatus will invariably contain (at least) a non-perturbatively suppressed source of uncertainty for any finite-mass apparatus. In gravity, the limit of decoupled infinitely massive local apparatus is obstructed — the system would collapse into a large black hole which would prevent probing short length scales, showing the operational meaning of UV/IR mixing. By the same token, global observables given by spacetime integrals of local quantities are also affected by uncertainties of this type.

**Asymptotic safety versus asymptotic darkness** — In this context, assuming weak (gravitational) coupling at least up to the EFT cutoff, various perturbative results based on the S-matrix bootstrap [272–284] point to stringy degrees of freedom: weakly coupled graviton scattering requires infinite towers of higher-spin degrees of freedom [272, 273, 285] to preserve

---

[24]As mentioned in section 3.1, perturbatively one can define observables that are local with respect to a gauge fixing, using e.g. a BRST procedure.

[25]Here we use the term in a possibly weaker sense than intended, e.g., in [97], where the existence of a *local* non-gravitational quantum system at the asymptotic boundary is required.

causality, while unitarity of the resulting amplitudes seems to only be compatible with a string spectrum and dynamics of the Virasoro-Shapiro type [128, 275–277] encountered in string theory. Even in the gauge sector, consistency of multiparticle factorization rules out at least some large families of deformations of the Veneziano amplitude which were previously not excluded [282]. However, while existing non-perturbative bounds on Wilson coefficients in the context of high-dimensional maximal supergravity [158, 159] are compatible with the ranges predicted by string theory, the case for the *necessity* of non-field-theoretic degrees of freedom is less strong. Since ASQG is a non-perturbative scenario, it may evade the above constraints on its S-matrix. However, due to its field-theoretic origin, the S-matrix of ASQG must exhibit (quantum) scale symmetry in the UV regime, namely it should feature power-law scaling. In particular, this means that (at least) a scattering amplitude should asymptote to a non-vanishing constant or power law at large center-of-mass energy $s \gg \Lambda_{\text{EFT}}^2$ (generically $s \gg M_{\text{Pl}}^2$) [286]. This behavior is in clear tension with expectations from *asymptotic darkness*, also known as black-hole dominance,[26] namely the idea that the high-energy spectrum of any theory of gravity be dominated by large black holes due to entropy bounds. As a result, in this scenario the high-energy behavior of gravitational amplitudes, at least in certain regimes of impact parameters, would be dominated by black-hole production [287–291] with an effective "single-particle" density of states in $d$-dimensional spacetime scaling as

$$\log \rho(E) \sim \left( \frac{E}{M_{\text{Pl}}} \right)^{\frac{d-2}{d-3}} \tag{4.1}$$

at high energies, invalidating a field-theoretic integral representation [254, 292]. This is because the existence of a Källén-Lehmann integral representation, or a canonical partition function, implies that the single-particle density of states for fields cannot scale super-exponentially. The arguments in [259] then translate the same bound to the multiparticle density of states. This argument actually dovetails with our preceding discussion on black-hole entropy in section 3.1, in the sense that both issues would be avoided if ASQG prescribed a different, non-thermodynamic interpretation to these quantities. Notably, such a scaling may be amenable to non-perturbative bootstrap methods in the future, which could further test whether an asymptotically safe field-theoretic S-matrix can be consistent with unitarity and causality principles. As a final remark, in asymptotically AdS sectors the above scaling changes, matching the one of a local conformal field theory on the boundary. However, asymptotic darkness would still make amplitudes decay faster than any power law.

To summarize, the above considerations on observables in QG seem to be consistent with the arguments involving the sum over topologies in the preceding section, leading again to the conclusion that holography and black-hole thermodynamics cannot appear in ASQG, at least not in the usual way. Perhaps, whatever regular effective geometry replaces black holes in such a scenario would shed light on what substitutes the thermodynamic interpretation of

---

[26]The super-exponential degeneracy of intermediate black-hole states implies that the amplitude for any particular final state be suppressed by the the reciprocal of this degeneracy. In particular, a two-to-two amplitude should decay faster than any power law in the (fixed-angle) limit of large center-of-mass energy.

black-hole entropy and its holographic area-like scaling in the semiclassical limit. Any such explanation would also need to explain why EFT computations involving horizons, carried out within its expected parametric regime of validity, ought to be reinterpreted differently from standard practice. In the next subsection, we discuss a feature of QG that appears to arise in certain limits in theory space and is grounded in information theory: the swampland distance conjecture (SDC). As we shall see, the ensuing considerations, when applicable, place further constraints on a consistent realization of ASQG, although they may be evaded whenever the theory space at stake is compact.

## 4.2 Infinite-distance limits in theory space

The SDC of [77] was first observed as a pattern in the string landscape. Concretely, the pattern that routinely appears when moduli are taken to infinity (in the relevant space) involves some degrees of freedom becoming light (in Planck units) and weakly coupled. Usually, such degrees of freedom drive the behavior of the theory in that regime, and in this context, they always form *infinite towers*. This feature is emphatically stringy and cannot be reproduced by quantum fields alone, and thus the SDC implies that QG is fundamentally different from a QFT. There are two simple ways to motivate why this behavior is markedly different from that of UV-complete quantum fields. To begin with, in field theory, geometric moduli associated with compact internal dimensions of space are accompanied by Kaluza-Klein towers. These only become light in certain limits, e.g., when the volume of the internal space becomes large in the Planck units of the lower-dimensional description. By contrast, in string theory winding modes and T-duality can be present.[27] Another way in which such infinite towers appear is via higher-spin excitations of weakly coupled strings, which can sometimes appear unexpectedly due to the intricate interplay of string dualities [294–296].

**Infinite towers from the bottom-up: factorization** — While the *universal* presence of such infinite towers seems to be particularly tied to string theory in nature, due to the above considerations, it is possible to provide a tantalizing, if incomplete, bottom-up rationale for these ideas. The starting point is the realization that the universal notion of distance in moduli spaces, and more generally theory spaces,[28] is provided by the (quantum) information metric, as reviewed in [88]. Families of probability distributions or quantum states are naturally equipped with a unique metric, and families of (effective field) theories similarly induce a quantum information metric via their ground states varying in their theory spaces or moduli spaces. Concretely, taking the absolute square of the inner product between neighboring states in the family defines the metric as the leading deviation from an exact overlap. This universal notion of geometry reduces to the commonly used ones whenever moduli are present, namely

---

[27]Even when no geometry is assumed, light towers of species arise and geometry emerges in these limits [156, 293].

[28]In a theory space we include moduli and discrete versions thereof (such as quantized fluxes), but also free (dimensionless) parameters such as couplings. In string theory the latter are not present, a fact which may be connected to swampland ideas [95]. In AdS/CFT it is also the case, regardless of string universality, up to the plausible identification of parameters with marginal deformations.

the conventional moduli space metric or the Zamolodchikov metric for (holographic) conformal field theories. The information metric encompasses them as special cases, and provides a physical and operational interpretation of singular points in terms of distinguishability by measurements [89].

According to the arguments in [89], whenever one takes limits in such spaces which lie at infinite distance with respect to this metric, observables must *factorize* in a suitable sense. In all known examples, this means that some (possibly composite or emergent) degrees of freedom become non-interacting in the limit, and their correlators factorize as products of one-point correlators. In order to obtain this result, unitarity is essential — a recurring theme in our analysis.

**Infinite towers and the equivalence principle** — Another recurring theme in this paper is that gravity is special, and its consistency is extremely and unexpectedly constraining from the EFT point of view; the properties of infinite-distance limits are no different. In this context, a key role is played by the equivalence principle, by which all degrees of freedom couple to gravity (universally at low energies). Therefore, factorization can never occur unless gravity itself decouples in the limit! Since the Planck scale is dimensionful, such a decoupling requires that the effective cutoff $\Lambda_{\mathrm{EFT}} \ll M_{\mathrm{Pl}}$ of the gravitational theory vanish in Planck units in the infinite-distance limit.[29] Put differently, the generic Wilson coefficient of the gravitational effective action[30] expressed in Planck units must diverge in the infinite distance limit. In [307–309], it was observed that this happens whenever a parametrically large number of species becomes light (see also [297, 310–314] for recent examples with specific Wilson coefficients). Although from the bottom-up perspective, the presence of such towers seems to be required for factorization in gravity [89] and perturbative UV completions of graviton scattering [272], it is not established that they must in fact be light, i.e. with a mass gap $m_{\mathrm{gap}} \ll M_{\mathrm{Pl}}$. At any rate, our ensuing considerations on infinite-distance limits do not rely on this latter requirement. From the AdS/CFT perspective, the analogous condition is the divergence of the central charge $C_T$, which need not occur at infinite distance in theory space. In fact, along a conformal manifold, $C_T$ remains constant,[31] and factorization is implemented in the bulk by the enormous gauge invariance provided by higher-spin towers [89].

**Infinite distance limits in ASQG** — It is unclear whether the presence of towers of species is the unique mechanism that implements factorization at infinite distances when gravity is present. If this were the case, one would be readily able to conclude that in ASQG, no infinite-distance limits are possible, except for decompactification limits in which extra compact dimensions become large. In this setting, the same issue would arise within the higher-

---

[29]An independent argument in this direction [297] stems from a dynamical analysis of end-of-the-world cobordisms [297–306]. Since we have discussed topology change in ASQG in section 3.1, we shall not consider these ideas in this section.

[30]More precisely, one can define redefinition-invariant Wilson coefficients, extracting them from the low-energy expansion of sharp observables.

[31]Moving across conformal manifolds varying $C_T$ and suitably adapting the information metric to discrete settings, one indeed finds light towers of species in concrete holographic examples [315, 316], while the exponential decay conjectured in [77] and motivated in [89] seems to reflect the presence of GR at low energies [317].

dimensional QFT until the maximal dimension is reached. This would imply, for instance, that any space of EFTs arising from ASQG is compact with respect to the quantum information metric, modulo decompactifications. While this conclusion does not rely on whether the tower be light, as advertised, the case in which it in fact is light brings along further implications. Since only limits in which Kaluza-Klein towers become light are allowed in field theory, any other limit would need to be obstructed by some mechanism, possibly a potential, along the lines of [318, 319]. However, even in such a scenario, the higher-dimensional theory obtained by decompactification could be re-compactified on a simpler manifold such as a circle, whose small-radius limit would then need to be obstructed, such as perhaps in [160, 161, 320]. Even in this scenario, at any rate, as already mentioned, one can take into account any possible decompactification until the maximal spacetime dimension of the field theory is reached (as we shall now tacitly assume without loss of generality), and the remaining theory space ought to be compact. In concrete ASQG settings, the maximal spacetime dimension is known from the outset, since it is part of the specification of the fundamental field theory, together with all its degrees of freedom. Hence, the issue of decompactification is immaterial in this sense, while the compactness of the remaining theory space follows from the presence of towers of species needed to factorize the information metric.

**Infinite distance limits without towers of species** — Alternatively, a different mechanism for factorization which does not involve such towers, light or otherwise, should be found. In such a case, infinite-distance limits would still involve a parametrically small cutoff in Planck units, in order to suppress gravitational couplings.[32] The latter statement is in principle testable in ASQG: even if a factorization mechanism without towers of species exists, the generic Wilson coefficient must diverge at infinite distance in theory space. A prototypical example may be that of the Goroff-Sagnotti coupling evaluated in a limit in which a gauge coupling $g$ vanishes, since the asymptotic information line element behaves as $\frac{dg^2}{g^2}$ [89, 152, 153].

Moreover, the magnetic version of the weak gravity conjecture [78] entails a specific parametric bound $\Lambda_{\text{EFT}} \lesssim g\, M_{\text{Pl}}$ for the cutoff, although its relation to magnetic monopoles and gauge completeness in ASQG would need to be clarified in light of our preceding kinematic considerations. The weak gravity conjecture [78, 86] is a sharp dynamical statement requiring the existence of sufficiently light particles charged under (continuous) Abelian gauge groups. Apart from its bottom-up motivations stemming from black-hole physics, the weak gravity conjecture also holds in perturbative string theory [321] and is well-supported in AdS/CFT holography [85, 151, 322], F-theory [294, 323] and from various positivity bounds in EFT [83, 84, 324, 325]. The weak gravity conjecture, at least in some averaged version, seems to be also connected to an $\mathcal{O}(1)$ breaking of the one-form center symmetry acting on Wilson and 't Hooft lines [87] and, magnetically, to the obstruction of recovering a global symmetry at zero coupling [78].[33]

---

[32]See [106] for a proposal of a specific scenario.

[33]In (extended) supergravity, these bounds imply the absence of parametric scale separation between the typical IR and UV scales (e.g.. the Hubble scale and the size of extra dimensions) [326–329], which seems to be

According to information-theoretic factorization, Wilson coefficients should also diverge in limits where the number of distinct quantum fields becomes large, such as in an $O(N)$ model in the large-$N$ limit. However, it is difficult to think physically of such a limit of a *bona fide* QFT, albeit at least formally defined and calculable. This intuition is corroborated in particularly restricted classes of EFTs such as supergravity in dimensions $d \geq 6$, where robust kinematic swampland constraints entail a finite number of light fields due to various anomaly constraints [101, 102, 166, 167, 169, 334].[34] Once again, such calculations can in principle be carried out in approximate settings.

**No infinite distance limits in ASQG?** — If the cutoff vanishes in Planck units along some infinite-distance limit, graviton scattering should become weakly coupled within its full regime of validity. Absent any additional light degrees of freedom, such as Kaluza-Klein towers arising from internal extra dimensions,[35] UV-completing such processes within the weakly coupled regime requires higher-spin towers [272], whose mass gap provides the relevant cutoff [128]. The gauge-theoretic counterpart of these processes leads to asymptotically Regge-like spectra [285], absent any limit point of the Coon type[36] which seem to be excluded in graviton scattering [275–277]. Moreover, the actual amplitudes seem to allow no deformation away from stringy predictions [275–277, 282]. As explained above, within strict ASQG no towers of species can arise up to (partially obstructed) decompactifications. Therefore, even the weaker version of the SDC phrased in terms of a cutoff would conflict with the notion of a field-theoretic description. A way out of these issues would be to preserve factorization without a weakly coupled regime arising in the limit, but the equivalence principle appears to forbid this [89]. Once more, to avoid inconsistencies of this type, we are led to the conclusion that, at least in their maximally dimensional sectors, ASQG theory spaces should not contain infinite-distance regions of theory space where the cutoff $\Lambda_{\text{EFT}} \ll \Lambda_{\text{QG}}$ is parametrically smaller than the scale at which the quantum coupling of gravitons becomes large.[37] In other words, if the arguments of [89] hold, no infinite-distance limits should arise except (possibly) for decompactification limits of internal extra dimensions, which lie in the field-theoretic framework. It would be interesting to study the geometry of possible ASQG theory spaces and the corresponding behavior of the EFT cutoff. Since non-perturbative gravitational UV physics is strongly coupled away from these limits, the simplest way for ASQG to be consistent with the above considerations would imply that e.g. couplings cannot be arbitrarily small. This condition *a priori* does not rule out "numerically small" couplings, such as the IR fine-structure constant $\alpha \approx 10^{-2}$ which may be considered small for some purposes.

---

another prominent feature of the string landscape [330] and holographic (super)conformal field theories [155, 331–333].

[34]Up to some sporadic infinite families which have not been excluded hitherto [335–338] (see, however, [339]).

[35]In [257, 259] it has been argued that this is the only possibility for such towers.

[36]Unlike Regge spectra, which are unbounded in mass, Coon spectra have a maximal mass to which the infinite tower of states asymptotes.

[37]As an example, in weakly coupled string theory the hierarchy takes the form $M_{\text{string}} \ll M_{\text{string}} \sqrt{\log g_s^{-1}}$ [257, 291, 340–342].

### 4.3 Summary and loopholes

In this section we provided arguments to support two main conclusions. The first, which connects to the kinematic considerations in section 3, is that field-theoretic degrees of freedom cannot reproduce the standard super-exponential density of black-hole microstates. In particular, high-energy scattering amplitudes in ASQG must have a scale-invariant high-energy behavior, which is incompatible with black-hole dominance, *a.k.a.* asymptotic darkness. The second conclusion is that infinite towers of light species cannot arise in all infinite-distance limits, and thus the information-theoretic factorization discussed in [89] must be achieved by other means, without infinite towers.

These ideas are based on two main ingredients. The first is the nature of observables in QG, in particular asymptotic observables. These are sharply defined in principle[38] and they avoid issues related to black-hole formation in measurements, gauge invariance and topology fluctuations. Insofar as the theory of QG at stake has superselection sectors with asymptotic boundaries, weakly coupled UV completions are significantly constrained by unitarity, (asymptotic/IR/quantum) causality and black-hole microstates. Such constraints may rule out some QFT-based weakly coupled QG theories. The second ingredient is information-theoretic factorization, which has been argued to arise at infinite distance via a suppression of the EFT cutoff of the gravitational sector. Even granting this premise, a potential way out is that factorization in the presence of gravity could occur without light towers of species.

All in all, some potential loopholes to the ideas and conclusions above are the following:

- since ASQG is strongly coupled, it avoids the constraints coming from perturbative bootstrap arguments;

- outside of weak coupling, addressing black-hole dominance and asymptotic darkness requires loopholes along the lines of section 3.5;

- no infinite-distance limits exist in ASQG theory spaces;

- information-theoretic factorization in the presence of gravity is not necessary at infinite distance;

- information-theoretic factorization occurs, but no infinite tower of species is needed to implement it.

It would be interesting to explore any of these scenarios, see also [106] for related discussions. We leave this question for future work.

## 5   Conclusions

In this paper, we thoroughly examined a number of lines of reasoning stemming from the swampland program and its conceptual foundations, in relation to the "strict" ASQG scenario

---

[38]Possibly up to infrared dressing [343–347].

(and other fundamental QFTs of gravity). We attempted to connect the dots from various kinematic and dynamical perspectives to compare the foundational principles of these ideas and their compatibility. As a result, we uncovered a structural source of tension between them, identifying topology change and black-hole thermodynamics as root causes.

On the QFT side, our arguments rely on the following three assumptions:

- a scenario of strict QFT: a Lorentz-invariant, unitary, causal and UV-complete quantum theory of finitely many fields;

- no strong IR non-localities (e.g. inverse box operators) at low energies — in other words, the gravitational IR is dominated by GR;

- EFT remains valid in the presence of horizons [136–138];

meaning that the conclusions below may be evaded by violating at least one of these points.

On the swampland side, we considered the network of swampland conjectures (see also figure 2) that are based on bottom-up arguments, i.e., independently of string theory or its patterns. These include the absence of global symmetries, completeness of gauge charges, their relations with black-hole thermodynamics and topology change, and the distance conjecture. The assumption here is that these are universal features of QG in the sense of [106].

To summarize our findings, the arguments that we have presented in this paper point to the following conclusions for fundamental QFTs of gravity:

- *The meaning of black-hole entropy cannot be ascribed to the usual framework of thermodynamics and statistical mechanics.*

  Although this was pointed out in [140], the arguments therein can be circumvented, e.g., by allowing for UV non-locality at the fixed point. In this paper, we reached the same conclusion as in [140] through a different (and stronger) chain of arguments, centered around the role of spacetime topology change in QG. In summary, we argue that fluctuations of spacetime topology — which underlie the most direct macroscopic derivations of black-hole entropy — are likely not realizable in gravitational QFTs *on spacetime*. The derivation of black-hole entropy based on matter fields would still work, indicating the existence of a(n almost) thermal bath, with which the fields are in equilibrium. The complementarity of these pictures, which can be seen as two sides of the same coin, would thus lead to a contradiction. Even if topology change were found to be consistent with gravitational QFTs, this would only fix the "macroscopic side" of the puzzle. A microscopic microstate counting is required and must match the macroscopic results. In this context, UV non-locality of the fixed point would be strictly necessary to possibly find an area scaling, but it might not be sufficient because of the corresponding "non-localizable" super-exponential density of states [239].

- *Modulo decompactifications, which would also require certain obstructions to shrinking limits, no other infinite-distance limits (at a fixed number of fields) should appear in the*

*theory space(s)*, unless a novel mechanism devoid of towers of species for information-theoretic factorization consistent with perturbative S-matrix bootstrap results be found.

These conclusions may be evaded by violating our starting assumptions, particularly by generalizing ASQG to either include infinitely many fields, or to a QFT that does not live on spacetime, e.g., a tensor model [249]. Alternatively, the first conclusion might be avoided by stipulating that black-hole microstates do not exist in ASQG. The second conclusion may furthermore be circumvented if the UV completion is strongly coupled (which is likely the case in ASQG), or if no infinite-distance limits exist in the ASQG theory space.

The deep connection between topology change, black-hole thermodynamics, and the "kinematic" swampland conjectures, is schematically summarized in figure 2. Our work thus identifies the first two as root causes of a potential conflict between the above network of swampland conditions and fundamental QFTs of gravity. A posteriori, they may be thought of as manifestations of UV/IR mixing expected in QG based on black-hole physics. The comparisons we drew with string theory and AdS/CFT highlight the existence of QG theories which are directly consistent with these swampland conditions. Namely, they are holographic and afford conventional interpretations of black-hole entropy in terms of microstates [215]. A scenario of effective ASQG would fall into this category of QGs as well, since topology fluctuations would be accounted for by the fundamental theory. The corresponding landscape would presumably be compatible with the swampland conjectures we discussed. By contrast, a consistent realization of fundamental ASQG (or similar QFT-based approaches) would constitute a qualitatively different type of QG in these respects. Nevertheless, the landscapes of low-energy EFTs could overlap as in figure 1. In summary, *either a fundamental description of gravity as a QFT (in the same dimension) is not possible, or the network of swampland constraints considered does not pertain to the universal landscape* [106] *(or both)*.

Besides such stringent constraints, what is left of UV consistency that can apply to ASQG? In other words, in the scenario in which the above sources of tension were eliminated, what would remain of such bottom-up constraints to help assess its consistency? One answer may be found in the non-perturbative S-matrix bootstrap, which may be able to test whether scattering amplitudes driven by quantum scale invariance are consistent with unitarity and (quantum) causality. Although such efforts have been mostly confined to low-energy constraints with maximal supersymmetry [158, 159], in principle this approach would complement direct computations to assess whether ASQG can be consistently realized as a QFT of the gravitational field.

## Acknowledgements

It is a pleasure to thank Leonardo Bersigotti, Jacques Distler, Dieter Lüst, Jacob McNamara, Miguel Montero, and Nicolò Risso for numerous insightful discussions, and Luca Buoninfante, Jacques Distler, Astrid Eichhorn, Miguel Montero, Jan M. Pawlowski, and Nicolò Risso for comments on the manuscript. I.B. and B.K. are grateful to Perimeter Institute for their

hospitality during the early stages of development of this work. The research of A.P. is supported by a research grant (VIL60819) from VILLUM FONDEN. A.P. and M.S. also acknowledge support by Perimeter Institute for Theoretical Physics during the development of this project. Research at Perimeter Institute is supported in part by the Government of Canada through the Department of Innovation, Science and Economic Development and by the Province of Ontario through the Ministry of Colleges and Universities. B.K. has been partially supported by Nordita. Nordita is supported in part by NordForsk. The work of I.B. was supported by the Origins Excellence Cluster and the German-Israel-Project (DIP) on Holography and the Swampland. The work of M.S. was in parts supported by a Radboud Excellence fellowship from Radboud University in Nijmegen, Netherlands.

## A    A dictionary for non-stringy readers

In this appendix we collect some definitions and explanations of technical terms which appear in the main text and which may be unfamiliar to the target audience of this paper.

- **Cobordism:** a notion of topological equivalence between manifolds. Two $n$-dimensional compact manifolds $M$, $N$ are said to be cobordant if there exists a manifold $W$ with boundary such that $\partial W = M \sqcup \overline{N}$, where for oriented manifolds the bar denotes the opposite orientation. More precisely, when considering manifolds with some (tangential) structure, such as orientation, (s)pin structure, or a map to some fixed space, one requires that a cobordism $W$ be equipped with the same kind of structure compatible with the restriction to its boundary. Denoting the tangential structure by $\xi$, one writes $M \sim N$ for the equivalence relation of cobordism, and the set of equivalence classes $[M]$ equipped with disjoint union form abelian groups $\Omega_n^\xi$ called bordism (or cobordism) groups. Bordism groups are useful in physics to encode a number of properties of topological phases of matter [348], anomalies [349] as well as topological transitions in QG [94].

- **Completeness of gauge charges:** the statement that in a gauge theory coupled to QG all possible gauge charges be present in the spectrum. Specifically, electric charges of states in $G$-gauge theory are labeled by finite-dimensional irreducible representations of $G$. The notion that the spectrum of gauge charges be complete amounts to saying that any such representation is carried by a state in the Hilbert space. It is motivated by the thermodynamics of charged black holes and by its relation with the absence of (generalized) symmetries [96, 97, 193, 194]. The former requires the existence of microstates for black holes solutions of any charge.

- **Endability:** a generalization of screening by particle-antiparticle pairs. Some operators supported on submanifolds can be defined on submanifolds with boundary, and are called endable. Usually, the boundary hosts a lower-dimensional operator — for instance, adjoint Wilson lines in gauge theory can end on field strengths. In electrodynamics, ending

Wilson lines requires charged matter, reproducing the familiar screening which gives rise to vacuum polarization. Endability of all such Wilson lines requires matter states with all possible charges in the spectrum. Endability is thus a notion of completeness in this sense, and in Yang-Mills theory it is equivalent to the absence of the one-form symmetry generated by Gukov-Witten operators (see [97], and references therein). In general this symmetry can be non-invertible (see below), but gravitational solitons break the non-invertible part [96] leaving its invertible center subsymmetry $Z(G)^{\text{1-form}}$ associated to the center $Z(G)$ of the gauge group $G$. The triviality of cobordism classes [94] implies endability of all operators [97].

- **ER = EPR:** the acronym for the slogan "Einstein-Rosen (bridge) = Einstein-Podolsky-Rosen (paradox)". The idea, first introduced in [350], is that quantum entanglement of states be dual to connectedness in spacetime [351] via wormholes. This information-theoretic perspective allows an independent derivation of black-hole entropy and entanglement entropy [352], as well as the encoding of a sum over topologies as quantum superpositions of factorized states [97]. Namely, a wormhole connecting two localized patches of boundary should be described holographically by a quantum superposition of factorized states, each of which is dual to a disconnected topology. By entangling in all possible ways one can generate any wormhole without topological obstructions [97]. See also [353] for a recent operational perspective, and [354] for recent developments on wormholes in the gravitational path integral.

- **Generalized and $p$-form symmetries:** symmetries in physical systems reflect the invariance of the dynamics under some transformations. As is well-known, they bring along a number of simplifications, such as conserved quantities, selection rules and breaking patterns. Generalized symmetries extend this notion in a way that retains such desirable properties, by recasting them in the language of topological operators. More concretely, ordinary symmetries in QFT are faithful unitary representations $G \to \mathcal{U}(\mathcal{H})$ of a group $G$ on the Hilbert space $\mathcal{H}$ which leave the energy-momentum tensor invariant. Equivalently, an element $g \in G$ is assigned to an operator $U_g(\Sigma_{d-1})$ supported on hypersurfaces of codimension 1 which is *topological*: any correlator $\langle U_g(\Sigma_{d-1})\dots\rangle$ is invariant under deformations (isotopy) of $\Sigma_{d-1}$ which do not cross any other insertion locus. For continuous symmetries, Noether currents $J^a$ generate $U_\alpha(\Sigma_{d-1}) \equiv \exp\left(i\alpha_a \int_{\Sigma_{d-1}} \star J^a\right)$ whose deformation invariance follows from local conservation $d\star J^a = 0$. This language suggests a generalization in terms of topological operators of any codimension: a $p$-form symmetry is obtained for topological operators supported on codimension $p+1$ submanifolds $\Sigma_{d-p-1}$, which act by linking on charged operators $\mathcal{O}(\gamma_p)$ supported on $p$-dimensional submanifolds [187]. If one further relaxes the condition that they arise from a group representation $U_g U_{g'} = U_{gg'}$, one can obtain more general "non-invertible" structures of the type $U_i U_j = \sum_k n_{ij}^k U_k$, encoded by (higher) fusion categories [188–192].

- **Holography:** a paradigm to understand QG in a picture where spacetime, quantum

fields and their interactions are effective, approximate notions which emerge from a different description. Several hints from semiclassical gravity suggest that the physics in a region of spacetime be encoded in its boundary [185, 186], quite counterintuitively for more familiar local physics of non-gravitational systems. Indeed, the gauge redundancies of gravity entail that the ADM Hamiltonian, when definable, is supported on the boundary of space, and no local observables exist. Furthermore, the area scaling of the entropy of large black holes is naturally reproduced by local boundary physics. We refer to this property in the main text as "boundary locality". A concrete framework to think about holography and match computations with GR at low energies is the AdS/CFT correspondence [149, 241], which beyond specific constructions in string theory can be also thought of as an axiomatic framework to define theories of QG with asymptotically AdS boundary conditions in terms of conformal field theory. Conversely, given a more general axiomatic framework for AdS sectors of QG, the question of whether any such theory is holographically dual to a conformal field theory is well-posed albeit unanswered.

- **Information geometry:** spaces of probability distributions come equipped with a natural Riemannian metric defined by Fisher, whose quantum counterpart for states is the (generalized) Fubini-Study metric (see [88, 89] and references therein for a review in the swampland context). Applying this notion to families of ground states, say of EFTs, one obtains a metric on theory space. In principle, such space can be parametrized by genuine (dimensionless and physical) free parameters, by moduli, or both. Points at infinity in this geometry are interesting, because they reflect an operational distinguishability between far separated points [88]. Furthermore, in such limits, correlators seemingly factorize, and correspondingly, some degrees of freedom decouple [89]. In gravitational EFTs, the equivalence principle seems to require infinitely many species to renormalize the cutoff down to a weakly coupled regime [89, 308, 309].

- **Species:** distinct types of quantum fields or particles. Because of the equivalence principle, the presence a large number of species in a theory of gravity suppresses the UV cutoff of the gravitational EFT. This can be shown e.g. by estimating loop corrections to perturbative gravitational quantities, such as the graviton propagator [307, 308, 355, 356]. Alternatively, this scale can be related to the size of the smallest black hole in the theory [256–258, 308, 357]. As a result, for $N$ species, the gravitational cutoff is (upper bounded by) the *species scale* $\Lambda_{\rm sp} = M_{\rm Pl}/N^{\frac{1}{d-2}}$. If the number of species grows unboundedly with mass, this definition becomes an implicit equation for $\Lambda_{\rm sp}$.

- **Vacua, landscapes, moduli and all that:** vacua are generically defined as states of lowest energy. In a theory of dynamical gravity it is not always clear how to define a suitable notion of energy, but there are favorable settings with timelike Killing vectors or fixed asymptotic boundary conditions which allow defining e.g. Komar, Bondi or Arnowitt-Deser-Misner masses. Although spacetime fluctuates in QG, fluctuations of

the asymptotic boundary are infinitely suppressed and define superselection sectors in the Hilbert space. These sectors are physically disconnected, in the sense that any matrix element between different ones vanishes. Within each sector, there can be semiclassical backgrounds which minimize an appropriate notion of energy. Since in gravity spacetime can be compactified, these backgrounds can yield several different types of low-energy physics, as discussed in [160, 161] in the context of the standard model EFT. Vacua can sometimes be connected by varying moduli, which in the EFT arise as expectation values of (gauge-invariant) scalar operators, resulting in moduli spaces. Similarly, discrete parameters such as quantized fluxes threading compact dimensions can label vacua.

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
