# Peer review of "Asymptotic safety, quantum gravity, and the swampland: a conceptual assessment"

_SciPost Physics_

## Round 3 · Referee Report · Anonymous (Referee 1) · 2025-12-5

Report

I acknowledge the authors' comments on my first report and the changes to the manuscript. I will reiterate that (i) breakdown of unitarity is a fundamental mathematical inconsistency from which now physical theory can hope to recover and (ii) the absence of diffeomorphism invariant local observables remains one of the most basic challenges for any field theory approach to quantum gravity. Notwithstanding, I am happy to recommend publication of this paper in SciPost.

Recommendation

Publish (easily meets expectations and criteria for this Journal; among top 50%)

---

## Round 3 · Referee Report · Anonymous (Referee 2) · 2025-12-9

Report

The authors have addressed most of my concerns and I appreciate their effort.

I emphasize that there should be a clear difference between saying that an approach does not account for topology change for practical reasons or if this is a foundational assumption.

Secondly, I would like to point out that the sentence written in their reply "More generally, the standard definitions of QFT take manifolds as input, there is no sum over manifolds (and any attempt of summing over manifolds in a path integral is problematic)" seems to contradict the statistical/combinatoric perspective from, e.g., matrix models.

These issues do not jeopardize any of the merits of the manuscript which I judge well-grounded and useful for a broad community. I recommend the work for publication.

Recommendation

Publish (meets expectations and criteria for this Journal)

---

## Round 3 · List of Changes

see replies to referee reports

---

## Editorial Decision

accepted_in_target_journal